# Integrated clinical and genomic analysis identifies driver events and molecular evolution of colitis-associated cancers

Walid K. Chatila [1,2,16], Henry Walch [2,16], Jaclyn F. Hechtman[3], Sydney M. Moyer [4], Valeria Sgambati[5], David M. Faleck[6], Amitabh Srivastava [3], Laura Tang[3], Jamal Benhamida [3], Dorina Ismailgeci [6], Carl Campos[7], Fan Wu [8], Qing Chang[9], Efsevia Vakiani [3], Elisa de Stanchina[5,9], Martin R. Weiser[8], Maria Widmar[8], Rhonda K. Yantiss[10], Manish A. Shah [11], Adam J. Bass [12], Zsofia K. Stadler[6], Lior H. Katz[13], Ingo K. Mellinghoff [7,14], Nilay S. Sethi [4], Nikolaus Schultz [2,7,15], Karuna Ganesh [5,6], David Kelsen [6] & Rona Yaeger [6] ✉

Inflammation has long been recognized to contribute to cancer development, particularly across the gastrointestinal tract. Patients with inflammatory bowel disease have an increased risk for bowel cancers, and it has been posited that a field of genetic changes may underlie this risk. Here, we define the clinical features, genomic landscape, and germline alterations in 174 patients with colitis-associated cancers and sequenced 29 synchronous or isolated dysplasia. *TP53* alterations, an early and highly recurrent event in colitis-associated cancers, occur in half of dysplasia, largely as convergent evolution of independent events. Wnt pathway alterations are infrequent, and our data suggest transcriptional rewiring away from Wnt. Sequencing of multiple dysplasia/cancer lesions from mouse models and patients demonstrates rare shared alterations between lesions. These findings suggest neoplastic bowel lesions developing in a background of inflammation experience lineage plasticity away from Wnt activation early during tumorigenesis and largely occur as genetically independent events.

Chronic inflammation has long been recognized to contribute to cancer development, particularly across the gastrointestinal tract. Colitis-associated cancers (CACs), which develop in patients with inflammatory bowel disease (IBD), exemplifies this connection, and CAC is considered the most serious complication of IBD. CAC provides a model to understand inflammation-mediated carcinogenesis: risk of CAC increases with longer duration of colitis, greater anatomic extent of colitis, and concomitant presence of other inflammatory manifestations[1]. Precancerous lesions developing in IBD, unlike sporadic adenomatous polyps, tend to be flat with less distinct borders and can be difficult to visualize[2]. Current guidelines

recommend that patients with a history of IBD proximal to the rectum of at least 8 years' duration undergo routine endoscopic screening for dysplasia every 1–5 years, depending on personal risk factors[3]. The incidence of CAC has decreased with regular surveillance, improved endoscopic techniques, and advances in IBD treatment[4,5], but risk remains 0.5–1% annually after 8 years of IBD[2] and about 20% lifetime.

Once diagnosed, CACs are treated the same as sporadic colorectal cancer (CRC). Clinical series suggest that patients with CAC have shorter survival than patients with sporadic CRC[6,7], and recent data suggest that CACs have distinct genomic features[8–10].

Our group and others have previously described recurrent genomic alterations and some of the distinct genomics of CAC[8–10]. In this study, we aimed to map out the genomic steps in the development of these lesions by sequencing dysplastic lesions and, where available, mucosa; evaluating the relationship between dysplasia and cancer or multiple primaries occurring in the same person; and expanding the genomic, genetic, and functional characterization of CAC.

Here, we define the clinical features, genomic landscape, and germline alterations in a large series of CAC. We find that neoplastic lesions in the setting of IBD predominantly emerge from independent genetic events, and there is no field of genomic changes predisposing to cancer development. These lesions do exhibit convergent evolution towards altered *TP53* and recurrent copy number changes and, unlike sporadic CRC, are largely independent of Wnt signaling, likely experiencing transcriptional rewiring away from Wnt signaling early during tumorigenesis.

## Results

### Study population

Overall, 174 patients with CAC were identified, consisting of 13 patients with full clinical data available, 131 patients with full clinical data and CAC specimens analyzed with next-generation sequencing, and 30 patients with only CAC next-generation sequencing results (Supplementary Fig. 1a). CAC cases in our initial series (*n* = 47)[9] were more fully annotated and included in this analysis. Full clinical and genomic data can be found in the Source Data file. Twenty IBD patients had dysplasia specimens analyzed with next-generation sequencing, including 16 patients with synchronous dysplasia and CAC and four patients with isolated dysplasia and no CAC. Eleven of the patients with dysplasia had their cancer specimen subjected to targeted sequencing and included in the "genomic cohort" (Supplementary Fig. 1b).

### Clinical features of CAC development

Characteristics of the CAC "clinical cohort" (*n* = 144) are summarized in Fig. 1a. Most patients were diagnosed after 10 years of IBD, but 13% (*n* = 18) were diagnosed after shorter IBD duration, including four patients diagnosed with IBD and cancer concurrently. At the time of CAC diagnosis, 76% of patients (*n* = 110) had active IBD under treatment or monitoring with a gastroenterologist, 3% (*n* = 5) developed cancer after total colectomy or restorative proctocolectomy in the anastomosis or stump, and 15% of patients (*n* = 22) had quiescent IBD off-treatment for many years and were no longer undergoing routine surveillance. Twenty-four patients (17%) had a documented prior colonoscopy within one year of CAC diagnosis. These data indicate that the risk for cancer persists if any colonic mucosa remains and clinical quiescence of IBD and routine endoscopic surveillance do not eliminate this risk.

At the time of CAC diagnosis, 62% of patients (*n* = 89) presented with worsening bowel symptoms (e.g., rectal bleeding, abdominal pain, obstruction); cancer was detected incidentally in an abscess, stricture, or fistula in 13% (*n* = 18); and 22% (*n* = 32) had cancer detected at surveillance colonoscopy. Surveillance colonoscopy identified all stages of disease; however, all cases of carcinoma in situ were detected by surveillance colonoscopy. Among patients where adjacent non-neoplastic mucosa was available for histologic review (*n* = 109), active enteritis/colitis was identified in 73% (*n* = 80), chronic inactive enteritis/colitis in 17% (*n* = 18), and the non-neoplastic bowel was histologically unremarkable in 10% (*n* = 11). Among the 89 CAC patients who developed metastases, most common metastatic sites were liver, peritoneum/abdominal wall/omentum, and lung; bone and pelvic metastases were significantly enriched in CAC compared to sporadic CRC[11] (Supplementary Fig. 2a–c).

### Genomic landscape of IBD-associated dysplasia and cancer

We performed next generation sequencing with targeted exon sequencing (166 CAC, 9 dysplasia) or whole-exome sequencing (WES)

**Table 1 | Clinical characteristics of sequenced colitis-associated cancer cases[a]**

| | | |
|---|---|---|
| IBD diagnosis | Ulcerative Colitis | 56.0% (*n* = 93) |
| | Crohn's disease | 44.0% (*n* = 73) |
| Duration of IBD at time of CAC diagnosis | <10 years | 13.3% (*n* = 22) |
| | ≥10 years | 84.9% (*n* = 141) |
| | Not available | 1.8% (*n* = 3) |
| Site of primary tumor | Small intestine | 10.3% (*n* = 17) |
| | Right colon | 27.1% (*n* = 45) |
| | Left colon | 29.5% (*n* = 49) |
| | Rectum | 31.3% (*n* = 52) |
| | Fistula tract | 1.8 % (*n* = 3) |
| Stage at diagnosis | 0 | 2.4% (*n* = 4) |
| | I | 16.9% (*n* = 28) |
| | II | 20.5% (*n* = 34) |
| | III | 28.3% (*n* = 47) |
| | IV | 29.5% (*n* = 49) |
| | Not available | 2.4% (*n* = 4) |
| Total number of CAC sequenced | | 166 |

[a]Data is shown at the tumor level.

(13 CAC, 20 dysplasia) to identify genomic alterations in IBD-associated dysplasia and cancer. Clinical characteristics of the "genomic cohort" of CAC patients are summarized in Table 1. *TP53* alterations were detected in about half of the dysplasia samples (Fig. 1b). Across dysplasia samples, multiple oncogenic drivers were seen, including *APC*, *KRAS*, and *SMAD4* mutations, with no alterations clearly emerging in the transition to carcinoma. In CAC, the most frequently altered genes were *TP53* (90%), *KRAS* (31%), *MYC* (20%), *APC* (20%), and *SMAD4* (13%). We compared the frequency of alterations in all recurrent genes in CAC by IBD subtype, and *PIK3CA* and *IDH1* alterations were significantly enriched in patients with a history of Crohn's disease versus ulcerative colitis. *KRAS* mutations were enriched in more differentiated CAC (Fig. 1c). The most frequent mutational signature, based on the subset of CAC with WES, was SBS1, which is associated with aging.

To evaluate for germline alterations in known cancer susceptibility genes that may contribute to CAC development, we reviewed germline sequencing results from the MSK-IMPACT germline panel in CAC patients who signed appropriate consent. Germline sequencing was available for 73 patients in the "genomic cohort" (45% of 161 patients) (Fig. 1d), and pathogenic alterations were identified in 10 patients (14%) (Supplementary Table 1). *APC* I1307K was most common (3 cases), likely reflecting the association of this alteration and of IBD with Ashkenazi Jewish ancestry. The *APC* I1307K variant, a unique low-penetrance germline alteration present in ~6% of patients of Ashkenazi Jewish ancestry, is not associated with Familial Adenomatous Polyposis (FAP) but rather with about a ~2-fold increased risk of CRC[12]. In itself, the *APC* I1307K mutation is not sufficient to cause cancer, but rather increases susceptibility of the *APC* gene to additional changes that may lead to colorectal cancer. *APC* germline mutations did not offset the low frequency of somatic *APC* alterations. Beyond the *APC* I1307K variant, several other germline alterations in genes involved in DNA repair were identified (*PMS2, ATM, RAD51B, DICER1, FANCA,* and *FANCC*). While germline alterations in *PMS2* are associated with Lynch syndrome, notably, both patients harbored microsatellite-stable CAC, suggesting that the *PMS2* alteration, as previously described, can be an incidental finding in some cancer patients[13]. Cancer susceptibility in patients with *ATM* and *RAD51B* alterations are still evolving with recent studies implicating these genes in a number of cancers; however, their contribution to CRC carcinogenesis at this time is unknown[14,15]. Similarly, while *DICER1* alterations have been linked to a variety of both benign

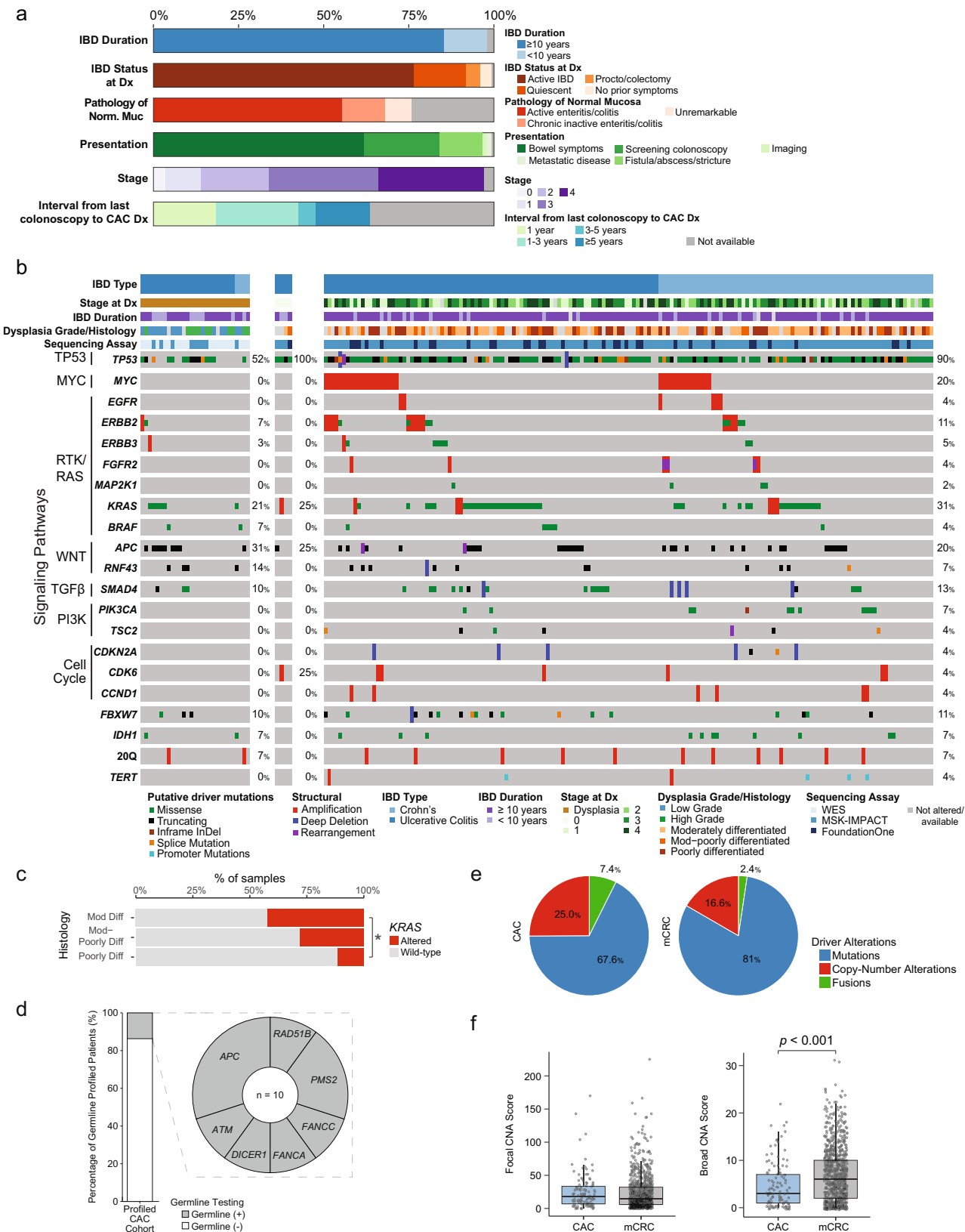

and malignant conditions, including colonic polyposis, the contribution of *DICER1* to CAC risk at this time is uncertain[16]. The monoallelic germline alterations in *FANCA* and *FANCC* reflect carrier status only; biallelic alterations in these genes are associated with the autosomal recessive condition of Fanconi anemia. Overall, we observed that the frequency of germline alterations was similar to that seen in CRC[17],

where an analysis of over 1000 patients using the same panel identified a prevalence of germline alterations of 14, 16, and 23%, in patients aged 50+ years, 36–49 years, and 14–35 years, respectively, with Lynch syndrome detected in 3, 4, and 8% of these groups, respectively.

Compared to sporadic, metastatic CRC[11], *MYC* and *KRAS* amplifications and *IDH1* mutations occurred more frequently in CAC, whereas

**Fig. 1 | Clinico-genomic characteristics of colitis-associated cancers. a** Clinical characteristics of 144 patients with CAC at time of CAC diagnosis. **b** Oncoprint showing recurrent genomic alterations in dysplasia ($n = 29$) and CAC ($n = 166$). Colors indicate type of genomic alteration and clinical characteristics as indicated in the legend below the oncoprint. **c** Frequency of *KRAS* alterations by CAC grade ($n = 115$). **d** Fraction of the patients who consented to germline testing ($n = 73$) who had germline alterations identified and the identity of these germline alterations. **e** Relative frequency of types of oncogenic changes in CAC and CRC ($n = 166$). **f** Frequency of focal and broad copy number alterations in CAC and CRC. Results

are shown for 1033 clinical IMPACT cases. The CAC ($n = 112$) and CRC ($n = 921$) groups were compared using a two-sided Mann−Whitney U-test and the significant results are as follows: CAC versus CRC broad CNA score, $P = 0.000016$. The center line of the boxplots indicates the median, the edges indicate the interquartile range, and the whiskers extend to the highest and lowest values not considered outliers. The asterisk shows when the comparisons were significant (* indicated significance here). Source data are provided as a Source Data file. Abbreviations: *Dx* diagnosis; *Norm* normal; *mCRC* metastatic colorectal cancer.

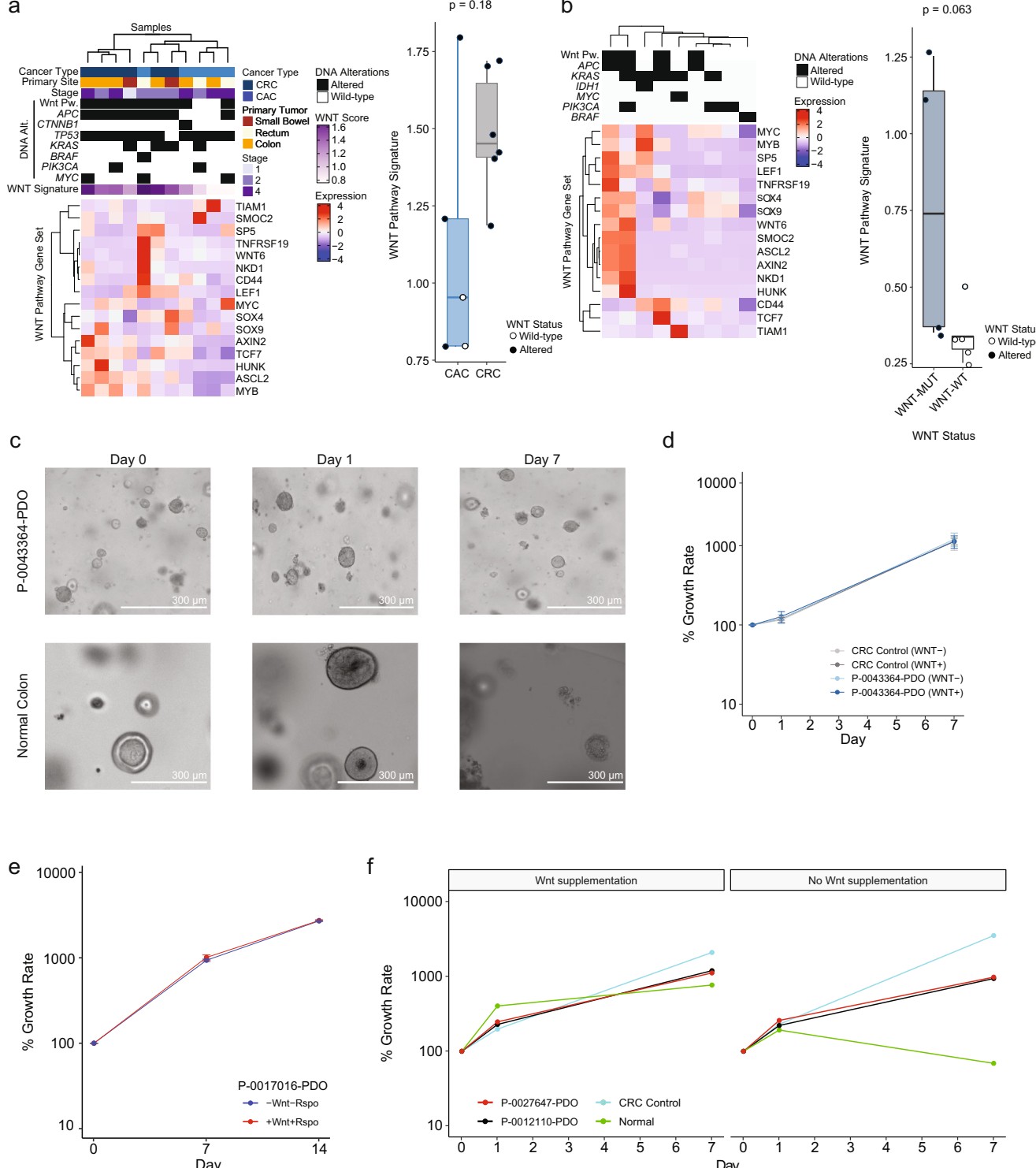

**Fig. 2 | Wnt signaling in colitis-associated cancers. a** Heatmap showing RNA sequencing results for Wnt pathway genes in CAC and CRC patient samples. Top panel indicates key genomic changes in the analyzed samples. Graph of Wnt pathway signature scores in CAC and CRC patient samples. In the box plot, results are shown for 11 patients: CAC ($n = 5$) and CRC ($n = 6$). **b** Heatmap showing RNA sequencing results for Wnt pathway genes in CAC patient-derived xenografts. Graph of Wnt pathway signature scores in CAC patient-derived xenograft samples by Wnt pathway status. In the box plot, results are shown for nine patient-derived xenografts: WNT-MUT ($n = 4$) and WNT-WT ($n = 5$). **c** Photographs of Wnt wild-type CAC organoid P-0043364 and normal colon organoid grown in conditions without Wnt supplementation. Experiments were conducted three times. **d** Growth curve for organoid P-0043364-PDO grown with and without Wnt supplementation (6 replicates each): CRC control, $P = 0.82$; P-

0043364-PDO, $P = 0.70$. The center is the mean growth and the error bars indicate standard deviations **e** Growth curve for organoid P-0017016-PDO grown with and without Wnt supplementation (6 replicates): $P = 0.82$. The center is the mean growth and the error bars indicate standard deviations. **f** Growth curve for CAC organoids (P-0012110-PDO, P-0027647-PDO) and normal bowel expanded in matrix with or without Wnt supplementation (6 replicates each): Normal, $P = 0.02$; colorectal cancer control, $P = 0.35$; P-0012110-PDO, $P = 0.41$; P-0027647-PDO, $P = 0.76$. In the box plots in a and b and for panels **d–f**, statistical significance was assessed using a two-sided Mann–Whitney $U$-test. The center line of the box plots indicates the median, edges indicate the interquartile range, and the whiskers extend to the highest and lowest values not considered outliers. Source data are provided as a Source Data file. WT wild-type, mut mutated.

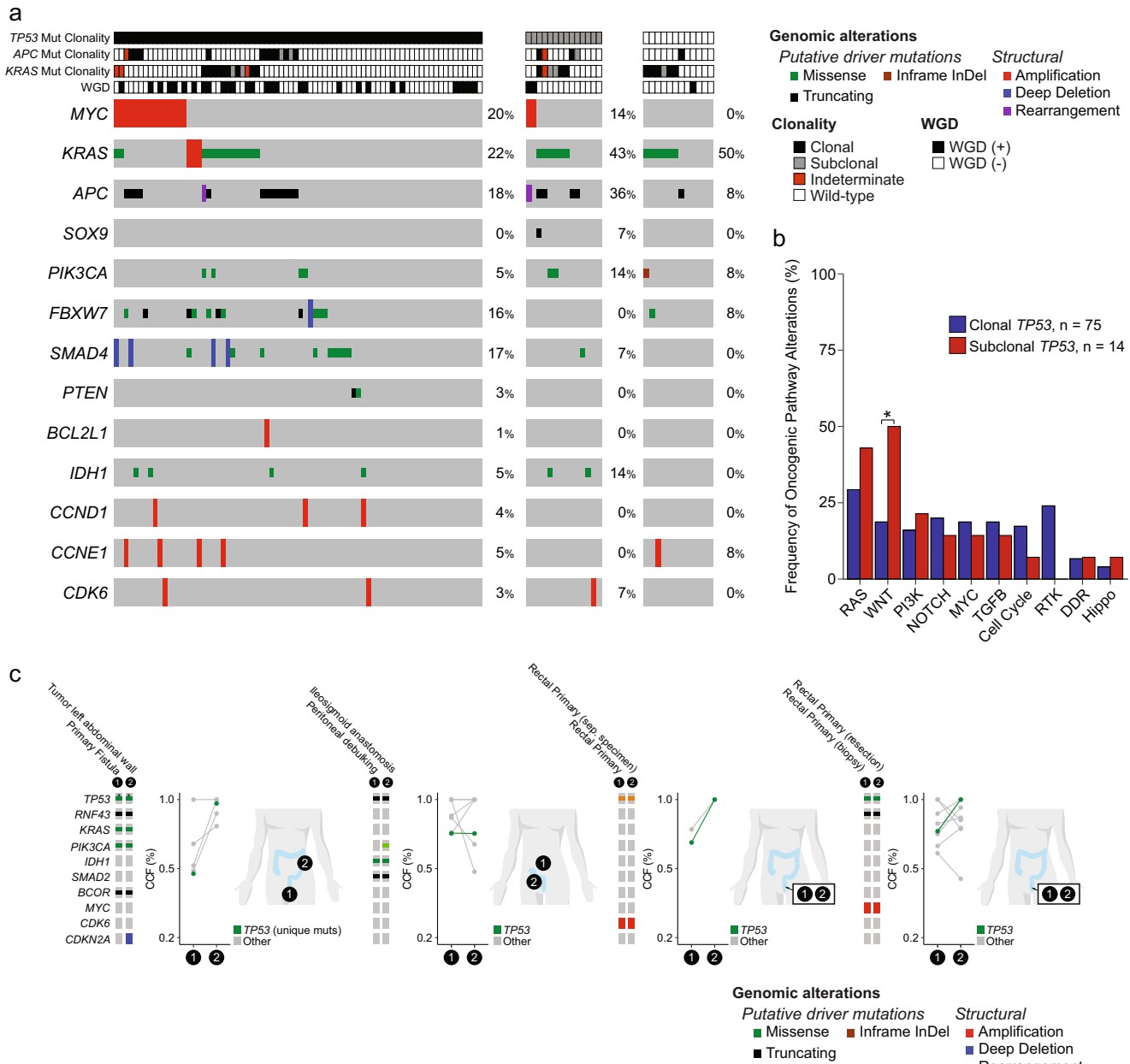

**Fig. 3 | _TP53_ clonality and concurrent alterations. a** Oncoprint of recurrent genomic changes by _TP53_ status (clonal, subclonal, or wild-type). **b** Frequency of pathway alterations by _TP53_ clonality. Asterisk indicates significant difference in comparisons performed with two-sided Fisher's exact test. The Wnt pathway was significantly enriched in the subclonal _TP53_ group versus the clonal _TP53_ group

($P = 0.017$). **c** Sites of multiple samples, genomic alterations, and changes in cancer cell fractions (CCF) of shared mutations from CAC specimens sequenced from patients with CAC with subclonal _TP53_ alterations. Source data are provided as a Source Data file.

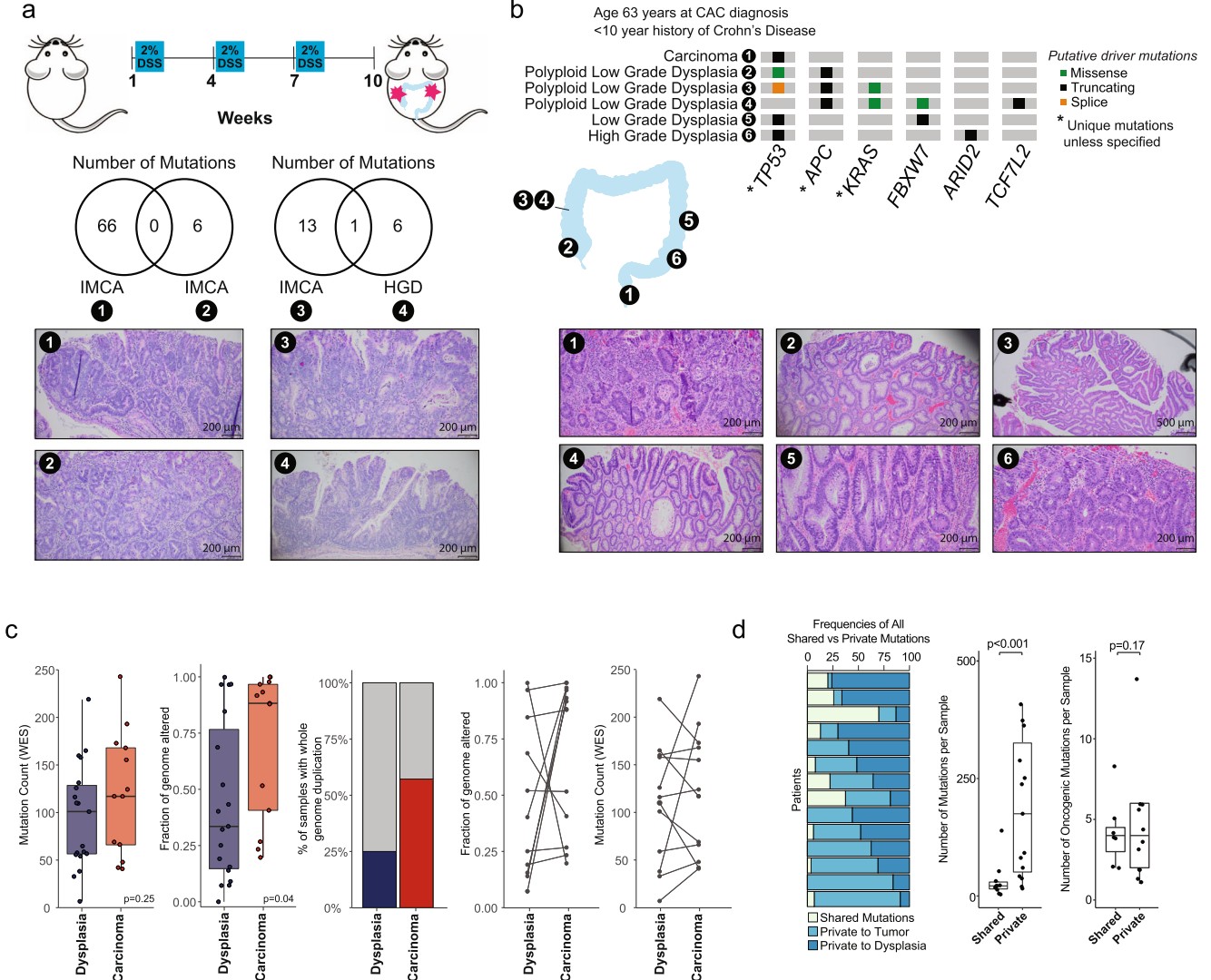

**Fig. 4 | Genomic characteristics of dysplasia arising in inflammatory bowel disease. a** Schema of AOM/DSS mouse model of inflammatory colon cancer, Venn diagram showing genomic alterations identified in two separate lesions genotyped from each mouse, and Hematoxylin & eosin stained slides showing areas macro-dissected for DNA extraction for sequencing. Experiments were conducted ten times and two representative mice were chosen for genomic analysis. **b** Genomic analysis of dysplastic/cancerous areas identified in a colectomy specimen, with illustrations showing locations of samples analyzed, Hematoxylin & eosin stained slides of areas macrodissected for DNA extraction, and oncogenic genomic alterations identified in each sample (filtered by OncoKB). Experiment was conducted one time. **c** Comparison of genomic changes detected by whole exome sequencing between dysplasia ($n = 19$) and CAC ($n = 12$). **d** Frequency of private and shared genomic alterations between patients with both a dysplasia and CAC sample ($n = 15$). In the left box plot, the number of shared and private mutations (all mutations) per pair of samples were compared ($P = 0.00022$). In the right box plot, the number of shared and private mutations (driver mutations only) per pair of samples were compared ($P = 0.17$). In the box plots in panels **c** and **d**, statistical significance was assessed using a two-sided Mann–Whitney U-test. The center line of the box plots indicates the median, edges indicate the interquartile range, and the whiskers extend to the highest and lowest values not considered outliers. Source data are provided as a Source Data file. AOM azoxymethane, DSS Dextran Sodium Sulfate, IMCA intramucosal carcinoma, HGD high grade dysplasia.

*APC, KRAS,* and *BRAF* V600E mutations occurred significantly less frequently (Supplementary Fig. 2d). After filtering by OncoKB[18], a larger portion of oncogenic alterations in CAC were copy number alterations compared to sporadic CRC (25.0% versus 16.6%) (Fig. 1e). Using copy number scores generated from CNapp (Supplementary Fig. 2e), we found that the frequency of focal copy number alterations was similar between CAC and sporadic CRC, however broad copy number changes were significantly more common in sporadic CRC ($P < 0.001$) (Fig. 1f); there were more focal amplifications of *MYC* in CAC (Supplementary Fig. 2f). In the 11 patients whose cancer developed in a background of histologically normal-appearing non-neoplastic mucosa, we found a higher frequency of *APC* mutations, suggesting that some of these may represent cancers that did not develop as a result of inflammation-induced neoplasia (Supplementary Fig. 2g). However, *MYC* amplification was frequently encountered in these tumors.

## Functional effect of genomic changes in CAC
*IDH1* R132 mutant tumors exhibited high global methylation consistent with the known functional effect of the oncometabolite 2-hydroxyglutarate (2-HG)[19,20], and we confirmed high 2-HG production in a patient-derived xenograft (PDX) generated from an ileal CAC (Supplementary Fig. 3a). We treated PDXs with selective pharmacologic inhibitors of IDH1 and FGFR alterations and confirmed tumor growth depended on these alterations (Supplementary Fig. 3b). These data identify potentially targetable alterations that are uniquely associated with CAC, distinct from sporadic CRC.

## Decreased Wnt signaling in CAC

Hyperactive Wnt signaling is thought to be a cardinal feature of intestinal cancers[21–23] and accordingly *APC* is the most common alteration in CRC[11,24]. In contrast, we identified Wnt pathway alterations in the minority of CAC; no alteration was specifically enriched in the *APC* wild-type CAC (Supplementary Fig. 4a); and neither *MYC* amplification nor *RNF43* mutation was selectively enriched in the wild-type cases. Beta-catenin immunohistochemistry performed in 14 CAC showed membranous staining in all Wnt pathway wild-type CAC (Supplementary Fig. 4b and Supplementary Table 2). RNA sequencing of tumors from five patients with CAC and six patients with sporadic CRC demonstrated a lower Wnt signaling ssGSEA score[25] in CAC compared to CRC ($p = 0.13$) and low Wnt gene expression in the Wnt pathway wild-type CAC (Fig. 2a). Consensus molecular subtype analysis showed no canonical CMS2 tumors in the CAC cases (Supplementary Fig. 4c), consistent with prior reports[10]. As our analysis was limited by the small number of available fresh tissue samples, we also examined RNA expression across our CAC PDXs, consisting of five Wnt wild-type and four *APC* mutant tumors (Fig. 2b). Only a subset of the *APC* mutant CAC exhibited elevated Wnt transcriptional output and the Wnt wild-type tumors exhibited low Wnt signaling ssGSEA scores. These data indicate that CAC without genetic alterations in the Wnt pathway have low Wnt pathway activity and have not activated this pathway through other, non-genomic mechanisms.

The low Wnt pathway activation in CAC suggests that these tumors may be largely independent of this pathway for growth, in contrast to normal bowel and sporadic CRC. To functionally evaluate this, we generated organoid models of CAC and used these models to probe CAC dependence on exogenous Wnt supplementation for growth. Wnt wild-type organoids (P-0043364-PDO and P-0017016-PDO) (Fig. 2c–e) and *APC*-mutant organoids (Fig. 2f) were able to grow without exogenous Wnt supplementation and their growth rate did not vary based on Wnt supplementation, in contrast to normal colonic organoids. Pharmacologic inhibition of Wnt ligand signaling with the porcupine inhibitor LGK-974 or of degradation of destruction complex components with the tankyrase inhibitor G007-LK in three CAC PDXs (two Wnt wild-type, one *APC*-mutant) did not suppress tumor growth (Supplementary Fig. 4d). RNA sequencing of two tumors from each condition (Supplementary Fig. 4e) indicated that tumors clustered by PDX model and not by treatment group. Further, the Wnt wild-type CAC had low Wnt ssGSEA scores. Together, these data suggest that the growth of CAC, including tumors without canonical Wnt pathway alterations, is independent of Wnt ligands from the tumor niche and that these tumors have undergone transcriptional rewiring to no longer depend on this pathway for growth.

## *TP53* alteration clonality and concurrent genomic changes

Prior molecular studies of CAC have demonstrated that *TP53* alterations can occur early and even be detected in normal-appearing mucosa[26–28]. Using the FACETS algorithm, we evaluated *TP53* clonality in CAC (Fig. 3a). The majority of CAC harbored clonal *TP53* alterations, but 14% of cases had subclonal *TP53* alterations and 12% had no detectable *TP53* alteration. Wnt pathway alterations were significantly enriched in CACs with subclonal *TP53* mutations (Fig. 3b). *IDH1* mutations were identified in CAC, regardless of *TP53* clonality. Analysis of paired samples from patients with subclonal *TP53* alterations, consisting either of a metastatic site or separate sampling of the primary tumor (Fig. 3c), indicated that a *TP53* mutation was always carried in both samples and often became clonal in the second specimen. All CAC with subclonal *TP53* and oncogenic copy number changes had an enrichment to clonal *TP53* alteration in the paired sample. Together, these data highlight the association of *TP53* alterations and copy number changes and whole genome duplication (WGD) in CAC. These data also suggest that *IDH1* mutations do not require early *TP53* mutation events but likely occur from a different milieu for CAC

development, an observation supported by the dysplastic lesion with an *IDH1* mutation without *TP53* alteration (Fig. 1b).

## Molecular steps underlying CAC development

To evaluate the relationship between multiple inflammation-associated malignant lesions and evaluate for a genetic field defect, we first analyzed tumors developing in the azoxymethane (AOM)/dextran sulfate sodium (DSS) mouse model of inflammatory colorectal cancer. This commonly used model of CAC uses chemical induction of DNA damage (via AOM administration) followed by repeated cycles of colitis (due to DSS application) to generate multiple neoplastic lesions over a period of weeks. Two mice were each treated with one dose of AOM and three cycles of DSS (Fig. 4a) until the development of multiple bowel lesions, 13 weeks in mouse 2609 and 11 weeks in mouse 3608. The resected bowel was then reviewed by an expert gastrointestinal pathologist and two separate lesions in close proximity (contained on one slide) were selected for macrodissection and mouse MSK-IMPACT sequencing. We found few shared genetic events between these close lesions: the intramucosal carcinoma and high-grade dysplasia analyzed from mouse 2609 shared a *CTNNB1* mutation and the two intramucosal carcinomas from mouse 3608 harbored no shared alterations.

Synchronous lesions are more commonly seen in patients with IBD compared with patients with sporadic CRC. In our clinical series, 110 patients underwent partial or total colectomy and 32% of these patients had synchronous, separate adenomas ($n = 7$), dysplasia ($n = 20$), or cancer ($n = 8$). In one patient undergoing total colectomy, we collected six synchronous lesions from across the bowel, ranging from low grade dysplasia to carcinoma (Fig. 4b). Sequencing showed no shared genetic alterations across these lesions. To further investigate the relationship between dysplasia and cancer, we analyzed 13 paired samples of dysplasia and CAC with WES (at 150× depth) (Supplementary Fig. 5a). Adjacent, normal-appearing mucosa was obtained from the 13 patients and also subjected to WES and blood was collected as a matched normal in 5 patients. Where blood was not available as a matched normal, the mucosa samples were used as a normal and manually reviewed for mutations in *TP53*, *APC*, and *KRAS*. We detected no *TP53* alterations in the mucosa samples. Pathway analysis (Supplementary Fig. 5a) indicated that copy number alterations of genes involved in the cell cycle were present in three carcinoma samples and not the synchronous dysplasia or any other dysplasia samples, suggesting that these changes may occur later in CAC development. Fraction genome altered and WGD were significantly increased in CAC versus dysplasia, consistent with ongoing genomic instability late in CAC development (Fig. 4c). Some of the dysplasia-carcinoma pairs shared genomic alterations and some had no shared genomic events (Supplementary Fig. 5b), but the majority of genomic alterations detected were private to the precancerous lesions or CACs (Fig. 4c).

We also performed targeted exon sequencing of separate primary tumors from six patients, two of whom had synchronous primary tumors and four with metachronous colorectal primaries (Fig. 5a, b). In all the patients, the multifocal tumors had no shared genomic changes, despite close proximities of the primary lesions in three of the patients. We saw evidence of convergent evolution in *TP53* alterations present across the cancers, but each lesion appeared to have developed independently at the genetic level. Altogether, our data indicate that the development of dysplasia and cancer, or of multiple cancerous lesions in IBD, does not require a shared early genetic step. In light of the low frequency of mutations identified in mucosa, the findings suggest that mutations in *TP53*, *KRAS*, and *APC* do not drive the field of vulnerability for CAC.

## Discussion

In this study, we have assembled the largest, annotated clinical and genomic series of CAC. We confirm our own and others' prior

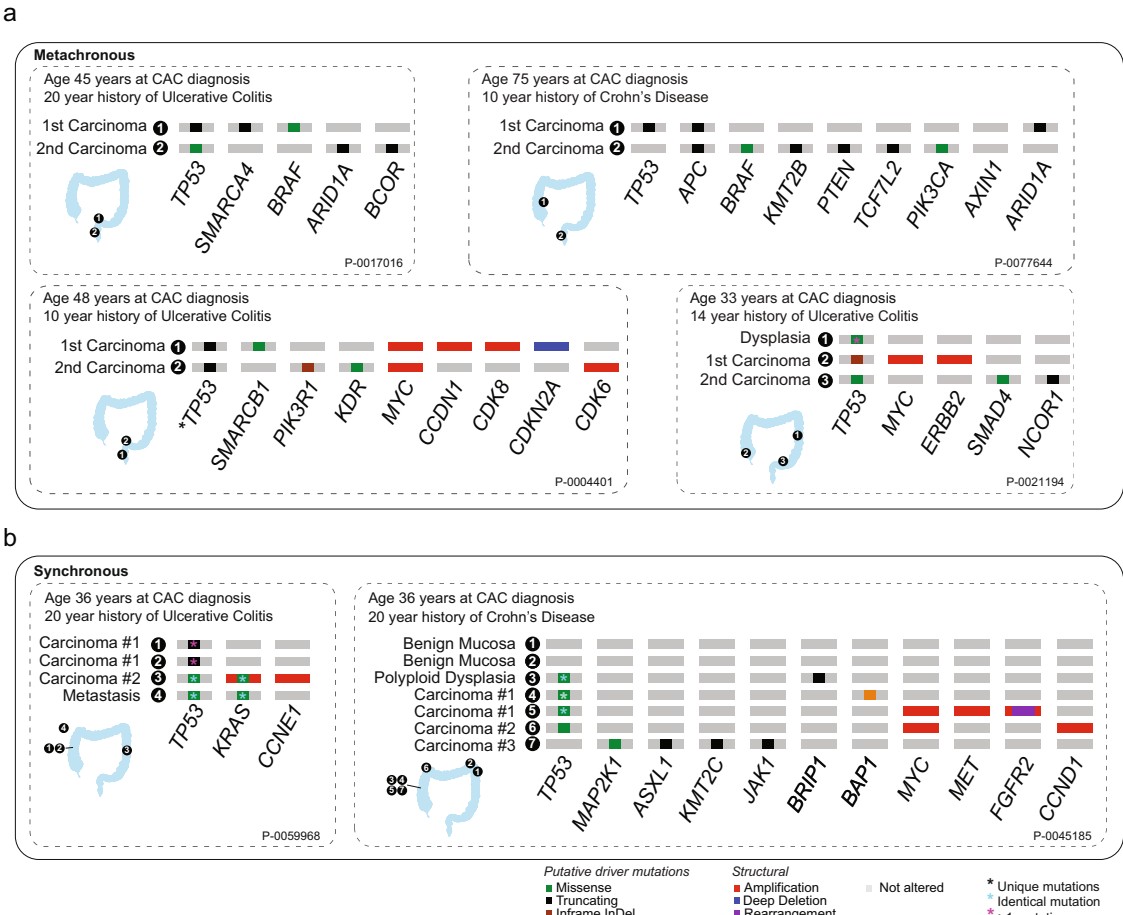

**Fig. 5 | Genetic relationship of multiple primary colitis-associated cancers.**
Genomic analysis of multiple primaries, with illustrations showing locations of
samples analyzed and oncogenic genomic alterations identified in each sample
(filtered by OncoKB). **a** Patients with metachronous primary colitis-associated
cancers: first row: (left) patient with rectosigmoid primary and 24 months later a
new primary in the rectal remnant and (right) patient with cecal primary and 4.5
years later a new sigmoid colon primary; second row: (left) patient with sigmoid
malignant stricture and 11 months later a new primary in the rectal stump and
(right) patient with cecal primary treated with right hemicolectomy, three years
later low-grade dysplasia in the distal transverse colon that was excised, and then
two and a half years later a sigmoid colon cancer treated with proctocolectomy.
**b** Patients with synchronous primary cancers: (left) patient with ascending and
sigmoid colon primaries and (right) patient with two distinct primaries in the
ascending colon and polypoid dysplasia with intramucosal cancer extending dis-
tally diffusely up to the splenic flexure.

identification of a distinct genomic landscape of CAC[8,9] and expand on
this to define copy number alterations in CAC, germline changes, and
functional effects of these genomic changes. Analysis of the clonality
of genomic changes, early lesions and mucosa, and multiple primaries
indicates substantial heterogeneity in CAC but the absence of a clear
genomic field effect for cancer risk. Using patient samples and patient-
derived organoids and xenografts, we demonstrate that unlike the
normal intestinal epithelium and sporadic CRC, which have high Wnt
signaling activity as a cardinal lineage defining feature, CAC are strik-
ingly independent of Wnt and seem to have undergone transcriptional
rewiring away from Wnt signaling and lineage plasticity early during
tumorigenesis[29].

Our work suggests that a different molecular pathway underlies
the development of CAC compared to sporadic CRC. Mutations may
not be the first event predisposing to cancer development in the
inflamed bowel. We failed to identify *TP53* alterations in mucosa and
only identified these mutations in about half of dysplasia samples. The
increasing frequency of *TP53* alterations from dysplasia to carcinoma is
in line with other series that suggest, in some CAC, *TP53* alterations
may occur later in the dysplasia-carcinoma transition[30]. Our analysis of
dysplasia and carcinoma that identifies late changes in the fraction
genome altered and WGD is also in agreement with prior studies

suggesting progressive accumulation of copy number alterations in
the transition from low-grade dysplasia to high-grade dysplasia and
carcinoma[31]. Further, analysis of multiple primaries showed that each
lesion developed independently at a genetic level, arguing against a
shared ancestral cell that undergoes early genomic changes leading to
cancer. It is, therefore, possible that non-genomic events, such as
epigenetic changes, may underlie the increased cancer risk seen with
inflammation. The transcriptional rewiring we identified leading to
Wnt independence may be one potential non-mutational early pre-
disposing event. We have previously shown that Lgr5 high/wnt high/
L1CAM low and Lgr5 low/wnt low/L1CAM high cells have distinct roles
with the former being important for homeostasis and sporadic CRC
tumor initiation and the latter for regeneration and metastasis[32]. Our
analysis supports a central role for such a Wnt-low population in CAC
initiation and our finding of low Wnt activation is in line with prior
evolutionary studies and transcriptional analyses of CAC[10,31,33].

We find that in immune-mediated bowel inflammation, neoplasia
predominantly emerges from independent genetic events, and there is
no field effect of genomic changes predisposing to cancer develop-
ment. Analysis of esophageal and pancreatic adenocarcinomas, where
inflammation has been implicated in cancer development, suggests
that when multiple neoplastic lesions are identified in these organs,

they can develop either as genetically-independent events or from a shared progenitor cell. In the esophagus, inflammation resulting from reflux of bile acids can lead to intestinalization of the lower mucosa, termed Barrett's esophagus, and elevated risk for esophageal adeno-carcinoma. An analysis of 25 pairs of separate areas of dysplastic Barrett's esophagus and esophageal adenocarcinoma indicated that the lesions were clonally unrelated in 11 cases and shared a common neoplastic clone in 14 of the cases[34]. Another study looking at 23 paired samples of Barrett's esophagus and esophageal adenocarcinoma found that the Barrett's esophagus was polyclonal even in the absence of dysplasia and that the spectrum of mutations in the paired specimens showed overall little overlap[35]. In 267 patients where sequencing was performed on samples collected about every 1–2 cm across the segment of Barrett's esophagus, a range of clonal diversity and divergence was identified with higher clonal diversity associated with higher chance of progression to cancer[36]. Analysis of pancreatic resection specimens containing at least one pancreatic intraepithelial neoplasia (PanIN) and adenocarcinoma present in anatomically distinct and far-removed regions, revealed that, in two of eight patients, the PanIN and adenocarcinoma were completely unrelated, in four patients, the PanIN and adenocarcinoma shared a common ancestor, and, in two patients, the two specimens shared all alterations[37]. Despite the differences in genomic alterations in multiple neoplastic lesions arising in the setting of IBD or other inflammation-associated cancers, the recurrent and distinct alterations in these lesions suggest a common pathway to neoplasia. It is possible that non-mutational changes underlie an early field defect in these lesions. For example, broad genomic copy number changes in Barrett's esophagus tissues and intestinal mucosa have been found to be highly predictive for progression to cancer[38,39].

Repeated cycles of inflammation, ulceration, and regeneration accelerate age-related mutational processes, and while chronic inflammation does not appear to be mutagenic per se, it may accelerate mutation accrual and provide a distinctive selective pressure[31,40]. In CAC, we observed a distinct genomic profile from sporadic CRC with a high frequency of *TP53* mutations and copy number changes. While rare in sporadic CRC, *FGFR2* amplification or fusion and *IDH1* R132 mutations occur in intrahepatic cholangiocarcinoma, another inflammation-associated gastrointestinal cancer, and, notably, these oncogenic drivers now have matched FDA-approved targeted therapies[41–43] supporting the potential of genomic analysis to alter the landscape of CAC treatment. The oncometabolite 2-HG has been shown to promote histone methylation, and *IDH1* mutations are associated with tumor hypermethylation phenotype in gliomas[19,20]. Interestingly, early series of CAC have noted tumor hypermethylation[44–46] and, in our analysis of global methylation, we see that five of the six *IDH1* mutant CACs clustered together with the microsatellite instable tumors and exhibited high global methylation, further supporting the functional significance of *IDH1* mutation in CAC.

Limitations of our study include the retrospective nature of sample collection and the relatively small number of dysplasia samples available. However, the large number of CAC samples and integrated clinical and genomic evaluation support our analysis of clinical and molecular features of CAC development. In the current study, we were able to perform matched tumor-normal sequencing for the majority of dysplasia and CAC cases studied, unlike previous studies from our group and others, allowing a better assessment of somatic alterations in these lesions. We performed primarily targeted sequencing of known cancer genes, with WES performed in a subset of cases, so the analysis may have missed important non-cancer genes involved in CAC development. Our data indicate that a substantial number of IBD patients develop cancer despite surveillance, and we hope that our identification of the specific genomics of CAC will support new approaches for early detection.

## Methods

### Samples

Samples were collected following ethical regulations. The analysis of CAC cases was approved by the Memorial Sloan Kettering Cancer Center (MSK) Institutional Review Board and received a waiver for informed consent (IRB waiver WA0143-14) under 45 CFR § 46.116 based on review and determination that this research meets the following requirements: (i) the research involves no more than minimal risk to subjects; (ii) the research could not practicably be carried out without the requested waiver; (iii) the waiver will not adversely affect the rights and welfare of the subjects. Biospecimens were collected under appropriate tumor banking protocols after patient written informed consent (IRB 06-107, 12-245, and 15-297). Germline analysis was performed in patients who provided informed consent (IRB 12-245 part C). Patients were not compensated for participation.

CAC cases were identified from Memorial Sloan Kettering Cancer Center ($n = 125$), Weill Cornell Medical Center-New York Presbyterian Hospital (WCMC) ($n = 29$), or Sheba Medical Center in Israel ($n = 7$). Samples were initially identified retrospectively through a query and validation of the pathology databases of these institutions for cases where colitis was a clinical factor noted in the pathology report, and CAC cases ($n = 47$) in our prior publication were included in this series[9]. Since 2015, cases have been identified prospectively at MSK through a biweekly computerized query for any patients with "colitis" as a diagnostic term who is scheduled to be seen in the surgical, gastroenterology, or medical oncology clinics. Pathology specimens were reviewed by expert gastrointestinal pathologists at MSK or WCMC. The medical records were reviewed by a medical oncologist for the clinical history of IBD. The pathologist selected the appropriate tissue blocks for DNA extraction. Patients with Lynch syndrome or microsatellite instable tumors were excluded to focus on molecular drivers related to inflammation. Areas of dysplasia were identified on standard hematoxylin-eosin stained slides for macrodissection.

### Clinical annotation

Pathology, colonoscopy, and radiology reports and medical oncology, surgery, and gastroenterology notes were reviewed in the electronic medical record to annotate clinical characteristics for patients with CAC.

### Genomic analysis

Samples were analyzed using the following hybrid capture-based next-generation sequencing assays: FoundationOne (315 genes, 34 samples)[9] and MSK-IMPACT (341-505 genes, 132 samples)[47] (Fig. 1b). Oncogenic or likely oncogenic somatic variants were identified using OncoKB, a precision oncology database that tracks the effects of cancer variants and their potential clinical actionability[12]. Genes were organized into signaling pathways using curated pathway templates[48]. FACETS, an allele-specific copy number algorithm was used to assess tumor purity, clonality of genomic alterations, and whole genome doubling events[49].

Sequencing of neoplastic lesions developing in the AOM/DSS mouse models was done with mouse MSK-IMPACT. Based on pathological review, 14 unstained slides per sample were macrodissected by scalpel to collect tumor material in an AutoLys M tube (Thermofisher Life catalog # A38738). FFPE curls collected in AutoLys M tubes (Thermofisher Life catalog # A38738) were digested with Protease Solution. DNA was extracted using the MagMAX FFPE DNA/RNA Ultra Kit (ThermoFisher catalog # A31881) on the KingFisher Flex Purification System (ThermoFisher) according to the manufacturer's protocol. Samples were eluted in $55\,\mu L$ elution solution. After PicoGreen quantification, 200 ng of mouse genomic DNA were used for library construction using the KAPA Hyper Prep Kit (Kapa Biosystems KK8504) with 8 cycles of PCR. After sample barcoding, 250–265 ng of each library were pooled and captured by hybridization with the

M-IMPACT_v2 (IDT) assay, which captures all protein-coding exons and select introns of 608 cancer-related genes (Supplementary Table 3). Capture pools were sequenced on the NovaSeq 6000, using the NovaSeq 6000 S4 Reagent Kit (200 Cycles) (Illumina) for PE100 reads. Following these criteria, the mean coverage was 623×, with an average of 99% of the targeted sequences covered 30×.

## Methodology for Copy Number Alterations (CNA) analysis

CNApp, a web-based tool, was used to analyze tumor sample segmentation data and generate both focal and broad CNA scores[50].

## Germline analysis

Germline sequencing was performed for patients who signed written consent for testing (MSK protocol 12-245 part C) and consisted of a panel of 76 genes in its first iteration (Version 1) and 88 genes in its second iteration (Version 2). The genes analyzed are listed in Stadler ZK et al.[51]. The overlap between Version 1 and Version 2 of the assay is 84% (14 genes were added and 2 removed for Version 2). Variants were identified by MSK-IMPACT as described above[52] and were interpreted based on American College of Medical Genetics and Genomics (ACMG) guidelines[53] by a clinical molecular geneticist or molecular pathologist. Changes from Version 1 to 2 reflect changes in the current understanding of genes involved in cancer susceptibility.

## Methylation analysis

Methylation profiling was performed using Methylation 450 K arrays (Illumina, San Diego, CA). DNA methylation data was processed in R using the *minfi* package. Beta-values were normalized (preprocessQuantile) and the function dmpFinder (option; "shrinkVar = TRUE, type = categorical"), which used an F-test, was used to identify differentially methylated positions between carcinomas-muc and their non-mucinous counterparts within each cancer type. Differentially methylated CpGs were identified using a cut-off of FDR < 0.05. Methylation probes were mapped to genes using the illuminaHumanMethylation450kanno.ilmn12.hg19 Bioconductor package[54].

## Immunohistochemistry (IHC)

Four-micrometer-thick sections were cut from formalin-fixed paraffin-embedded (FFPE) tumor blocks for IHC. IHC for β-catenin was performed on a BenchMark XT automated immunostainer (Ventana Medical Systems Inc., Tucson, Ariz). Sections were incubated with anti-β-catenin antibody (Cell Marque, catalog #760-4242) at a concentration of 1.73 μg/mL. Antigen retrieval was performed with Cell Conditioning Solution (CC1, Ventana Medical Systems Inc.) for 24 h, and primary antibody incubation was for 24 h. Antigen detection was performed using the Optiview DAB Detection kit (Ventana Medical Systems Inc.).

## RNA sequencing and analysis

Frozen tumor tissue from clinical samples or patient-derived xenografts (PDXs) was subjected to RNA sequencing. 20–30 mg frozen tissue was homogenized in 1 mL TRIzol Reagent (ThermoFisher catalog # 15596018), phase separation was induced with 200 μL chloroform, and RNA was extracted from the aqueous phase using the miRNeasy Micro Kit (Qiagen catalog # 217084) on the QIAcube Connect (Qiagen) according to the manufacturer's protocol with 350 μL input. Samples were eluted in 15 μL RNase-free water. After RiboGreen quantification and quality control by Agilent BioAnalyzer, 500 ng of total RNA with RIN values of 9.2–10 underwent polyA selection and TruSeq library preparation according to instructions provided by Illumina (TruSeq Stranded mRNA LT Kit, catalog # RS-122-2102), with 8 cycles of PCR. Samples were barcoded and run on a HiSeq 4000 in a PE50 run, using the HiSeq 3000/4000 SBS Kit (Illumina). An average of 44 million paired reads was generated per sample. Ribosomal reads represented 1.4–3.0% of the total reads generated and the percent of mRNA bases averaged 67%. Differential gene expression analysis was conducted using DESeq2 v.1.30.165. The biomaRt v.2.46.3 package66 was used to annotate genes. Single sample gene set enrichment analysis (ssGSEA) was performed using the R package GSVA 67.

## Generation of patient-derived organoids and xenograft models and growth curve experiments

Patient-derived organoids were generated under MSK IRB protocols 06-107 and 14-244. Normal colon crypts were isolated by agitating with 8 mM EDTA in PBS. Human tumor samples were grossly washed, chopped into 5 mm fragments, and incubated in dissociation buffer (DMEM with 5% FBS (Gibco), 2 mM l-glutamine (Fisher Scientific), penicillin-streptomycin (Fisher Scientific), 40 μg ml$^{-1}$ gentamicin. (Thermo Fisher Scientific), 250 U ml$^{-1}$ type III collagenase (Worthington) and 1 U ml$^{-1}$ dispase (Sigma-Aldrich)) on a shaker for 30 min at 37 °C, filtered through a 70 μm cell strainer (Greiner Bio-One), centrifuged at $600 \times g$ for 5 min and washed with ADF (Advanced DMEM/F12, Thermo Fisher Scientific). Cells were counted and resuspended in Matrigel at approximately 2000–10,000 cells per 40 μl of Matrigel in uncoated CELLSTAR multiwell culture plates (Greiner Bio-One). After Matrigel polymerization, human intestinal stem cell ((HISC): Advanced DMEM/F12 containing 100 ng ml$^{-1}$ Wnt-3a (R&D), 1 μg ml$^{-1}$ R-Spondin1 (Peprotech), 50 ng ml$^{-1}$ EGF, 50 ng ml$^{-1}$ Noggin (Peprotech), 10 nM gastrin (Sigma), 10 mM nicotinamide (Sigma), 500 nM A8301 (Sigma), 10 μM SB202190, 10 mM HEPES, 2 mM glutamine, 2 mM N-acetylcysteine, 1 μM PGE2 (Sigma), 1:100 N2 (Invitrogen), 1:50 B27 (without vitamin A) and 100 μg ml$^{-1}$ Primocin (InvivoGen)) was added. Y27632 (10 μM; Sigma) was added for initial organoid generation and for 48 h after every passage. Where indicated, cells were flow-sorted before resuspension and plating in Matrigel. For experiments, organoids were cultured in HISC with or without 100 ng ml$^{-1}$ Wnt-3a (R&D), 1 μg ml$^{-1}$ R-Spondin1 (Peprotech). Tumor growth was monitored using Cell-Titer glo assays using the manufacturer's protocols (Promega).

PDX models were generated by mincing about 1 g of tumor tissue, mixing with matrigel (50%), and implanting orthotopically into NSG (NOD scid gamma) mice (JAX strain 005557) (IRB protocols 06-107, 14-091). The growing tumor was then implanted as subcutaneous xenografts for growth experiments, and three-dimensional tumor measurements were performed. PDX experiments were performed using 4–6-week-old NSG female mice and five mice were included per treatment group. Treatment of the mice began when tumors reached approximately 100 mm$^3$ in size. Mice were randomized ($n = 5$ mice per group) to receive drug treatments or vehicle as control. AGI-5198 (50 mg/kg) was given twice daily by oral gavage. Debio1347 (60 mg/kg) was given daily by oral gavage. G007-LK (30 mg/kg) was given by injection once daily. LGK-974 (3 mg/kg) was given daily by oral lavage. The average tumor diameter (two perpendicular axes of the tumor were measured) was measured in control and treated groups using a caliper. The data are expressed as the increase or decrease in tumor volume in mm$^3$ (mm$^3$ = π/6 × (larger diameter × (smaller diameter)$^2$)). Investigators were not blinded when assessing the outcome of the in vivo experiments. These studies were performed in compliance with MSK institutional guidelines under an Institutional Animal Care and Use Committee (IACUC) approved protocol. The maximal tumor size was not exceeded, and the animals were immediately euthanized as soon as the tumors reached the MSK IACUC set limitations (1500–2000 mm$^3$).

Organoid and PDX samples were subjected to MSK-IMPACT sequencing and compared to clinical sequencing of diagnostic tumor tissue to validate the conservation of genomic alterations between these models and the tumors from which they were derived. Organoid samples were collected for sequencing after at least 6 months growth in culture and PDX samples were collected once xenografts were growing exponentially. Patient-derived organoids and xenografts can be requested from the corresponding author and will be shared, subject to institutional approvals.

## AOM/DSS Mouse Model of CAC

Male wild-type C57BL/6J mice (Jackson Laboratories stock no: 000664) ($n = 10$) were administered a single intraperitoneal injection of 10 mg/kg Azoxymethane (AOM, Sigma-Aldrich A5486) at 8-to-10 weeks of age. Following AOM injection, mice immediately began cycles of 2% dextran sulfate sodium (DSS, MP Biomedicals 0216011080) dissolved in RO drinking water (w/v). Mice were administered 2% DSS water for 7 days, followed by 14 days of normal RO drinking water (1 DSS cycle). Mice were administered a total of 3 DSS cycles spanning a 9-week period. Mice were euthanized 10–13 weeks after AOM injection. Two mice were selected for the collection of neoplastic lesions and sequencing. All animal procedures were approved by the IACUC of Dana-Farber Cancer Institute (protocol 11-009).

Following euthanasia, mouse small intestine and colon were removed and flushed with PBS. Tissue was then fixed in 10% neutral buffered formalin for 18–24 h at room temperature. Following fixation, intestines were opened longitudinally, 'swiss-rolled', and placed in a tissue cassette. Cassettes were placed in 70% ethanol for an additional 24 h prior to processing for paraffin embedding. Tissue processing, paraffin embedding, 5 μm sectioning, and Hematoxylin and Eosin (H&E) staining were performed by the Harvard Digestive Diseases Center Microscopy and Histopathology core located at the Beth Israel Deaconess Medical Center (BIDMC).

## 2-HG measurement

2-HG quantification was performed in brain and tumor tissues by Agios using their bioanalytical protocol[55]. Cell pellets were extracted with MeOH:H$_2$O at 80:20 ratio and compared to 2-HG standard to calculate 2-HG levels.

## Drugs

AGI-5198 was provided by Agios. Debio1347 was obtained from Debio-Pharm. G007-LK and LGK-974 were purchased from Selleckchem.

## Statistical analysis

Clinical characteristics and genomic frequencies were compared using a 2-sided Fisher's exact test. Continuous variables were compared using a Mann–Whitney U-test. Cumulative incidence rates were compared using Gray's test. Multiple testing correction was performed using the Benjamini–Hochberg method and significance was determined using a $q$-value cutoff of 0.05. Growth curve raw values were normalized relative to start date values and statistical significance was assessed comparing these values at the start date to the values at the end date using a Mann–Whitney U-test. The R statistical software (3.6.1) was used for analyses.

## Reporting summary

Further information on research design is available in the Nature Portfolio Reporting Summary linked to this article.

## Data availability

All clinical and genomic sequencing data described in this manuscript have been deposited in the cBioPortal for Cancer Genomics and are publicly available for online browsing and bulk download through the following link: http://www.cbioportal.org/study/summary?id=bowel_colitis_msk_2022. The raw sequencing data are protected, and patient consent to deposit raw sequencing data was not obtained. De-identified data are available under restricted access to protect patient privacy in accordance with federal and state law. These data can be requested for research use from the corresponding author, subject to institutional approvals. Additionally, all clinical data and variant call data are included in the Source Data file. The raw RNA sequencing data generated in this study have been deposited in GEO (accession number GSE220067). All other data generated in this study

are available within the article and its supplementary data files. Source data are provided with this paper.

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

## Acknowledgements

This work was supported by a Starr Cancer Consortium grant (I11-0040) and a Crohn's and Colitis Foundation Senior Researcher Award (#426095). We acknowledge the use of the Integrated Genomics Operation Core, funded by the NCI Cancer Center Support Grant (CCSG, P30 CA08748), Cycle for Survival, and the Marie-Josée and Henry R. Kravis Center for Molecular Oncology. We thank Matthew Vogel for his review and suggestions.

## Author contributions

Conception and design: W.K.C., H.W., N.S.S., K.G., D.K., and R.Y.; acquisition, analysis, or interpretation of data: all authors (W.K.C., H.W., J.F.H., S.M.M., V.S., D.M.F., A.S., L.T., J.B., D.I., C.C., F.W., Q.C., E.V., E.dS., M.R.W., M.W., R.K.Y., M.A.S., A.J.B., Z.K.S., L.H.K., I.K.M., N.S.S., N.S., K.G., D.K., and R.Y.); writing of the first manuscript draft: R.Y.; substantial revision of the first manuscript draft: W.K.C., H.W., D.M.F., N.S.S., N.S., K.G., D.K., and R.Y.; approval of the submitted manuscript: all authors (W.K.C., H.W., J.F.H., S.M.M., V.S., D.M.F., A.S., L.T., J.B., D.I., C.C., F.W., Q.C., E.V., E.dS., M.R.W., M.W., R.K.Y., M.A.S., A.J.B., Z.K.S., L.H.K., I.K.M., N.S.S., N.S., K.G., D.K., and R.Y.); agreement to be personally accountable for the author's own contributions and to ensure that questions related to the accuracy or integrity of any part of the work, even ones in which the author was not personally involved, are appropriately investigated, resolved, and the resolution documented in the literature: all authors (W.K.C., H.W., J.F.H., S.M.M., V.S., D.M.F., A.S., L.T., J.B., D.I., C.C., F.W., Q.C., E.V., E.dS., M.R.W., M.W., R.K.Y., M.A.S., A.J.B., Z.K.S., L.H.K., I.K.M., N.S.S., N.S., K.G., D.K., and R.Y.).

## Competing interests

N.S.: Advisory role for Astrin Biosciences. Z.K.S.: Immediate family member serves as a consultant in Ophthalmology for Alcon, Adverum, Gyroscope Therapeutics Limited, Neurogene, and RegenexBio, outside the submitted work. I.K.M.: Research funding from General Electric, Agios, Amgen, and Lilly; advisory roles with Agios, Debiopharm Group, Puma Biotechnology, and Voyager Therapeutics; and honoraria from Roche for a presentation. R.Y.: Research funding from Boehringer Ingelheim, Mirati Therapeutics, Pfizer, and Daiichi Sankyo; advisory roles with Mirati Therapeutics, Natera, Array BioPharma/Pfizer, and Zai Lab. All other authors declare no competing interests.

## Additional information

[1]Tri-Institutional Program in Computational Biology and Medicine, Weill Cornell Medicine, New York, NY, USA. [2]Marie-Josée and Henry R. Kravis Center for Molecular Oncology, Memorial Sloan Kettering Cancer Center, New York, NY, USA. [3]Department of Pathology, Memorial Sloan Kettering Cancer Center, New York, NY, USA. [4]Department of Medical Oncology, Dana-Farber Cancer Institute, Boston, MA, USA. [5]Molecular Pharmacology Program, Memorial Sloan Kettering Cancer Center, New York, NY, USA. [6]Department of Medicine, Memorial Sloan Kettering Cancer Center, New York, NY, USA. [7]Human Oncology and Pathogenesis Program, Memorial Sloan Kettering Cancer Center, New York, NY, USA. [8]Department of Surgery, Memorial Sloan Kettering Cancer Center, New York, NY, USA. [9]Antitumor Assessment Core Facility, Memorial Sloan Kettering Cancer Center, New York, NY, USA. [10]Department of Pathology, Weill Cornell Medicine, New York, NY, USA. [11]Department of Medicine, Weill Cornell Medicine, New York, NY, USA. [12]Herbert Irving Comprehensive Cancer Research Center, Columbia University Irving Medical Center, New York, NY, USA. [13]Gastroenterology Institute, Sheba Medical Center, Tel Hashomer, Ramat Gan, Israel. [14]Department of Neurology, Memorial Sloan Kettering Cancer Center, New York, NY, USA. [15]Department of Epidemiology-Biostatistics, Memorial Sloan Kettering Cancer Center, New York, NY, USA. [16]These authors contributed equally: Walid K. Chatila, Henry Walch. ✉e-mail: yaegerr@mskcc.org

