## [Peer Review File · Nature Communications]

REVIEWER COMMENTS

Reviewer #1 (Remarks to the Author): Expert in colorectal cancer functional genomics, organoids, and mouse models

The manuscript by Chatila et al. contribute to understand how chronic inflammation influences colon cancer development. They combine clinical data, genomic, and germline alterations to mainly suggest that inflammatory conditions promote lesions that occur as genetically independent events. The manuscript is interesting however I find that the data presented lacks certain novelty. I find it is an extended report of what has been published in the topic by the same authors and/or other colleagues in the field. For example:

Robles et al., *Gastroenterology* 2016 (whole-exome seq IBD; less number of cases but overlapping findings with TP53, APC and distinct genetic features of CACs compared to sporadic cancers)

Yaeger et al., *Gastroenterology* 2016 (same group and analyzed data collected until 2016, focussed on TP53, IDH1, genetic features and CD/UC differences)

Rajamaki et al., *Gastroenterology* 2021 (CACs and Wnt pathway, genetic and epigenetic events)

These reports are briefly explained throughout the text in spite of its relevance.

The data: well analysed, well interpreted and conclusions describe the main findings.

The methodology: well described and it is obvious from the report that the group has a strong background in these methodologies.

Major points

1. The authors should make clear what is the novelty compared to previous studies and in the manuscript state what are the samples that were used in the 2016 manuscript. Also, I would have expected a brief paragraph in the introduction to explain the new aim of this study and the novelty compared to the previous one.

1. Why do the authors not compare UC and CD data? They did not find major differences? and only the ones reported in Yaeger et al 2016. Or alternatively, are these previous findings confirmed?

2. Figure 1 and 2 lacks novelty. However, Figure 3 data are interesting. Can the authors extend the analysis testing PDOs from these samples and drug testing? or alternatively find a way to analyse further the distinct genetic profiles? can they present statistics to proof well the findings?

3. One of the main challenges is to proof that CACs can be treated according to its signature/profile and that this would be more efficient than when CACs are treated as sporadic tumours. The authors should perform some experiments in this respect. This would give the novelty that is lacking.

Minor points:

1. Hstology in Figure 5a should be improved (magnification, arrows showing the major differences...)

2. Abstract should describe better the novelty of the study.

3. In general, the manuscript statistics could be improved in the methods section and the p values description. P values should be clear in figures.

4. Genomic analyses were done by FoundationOne in 34 samples and MSK-IMPACT in 132 samples. These methods only reveal hundreds of genes and the limitation should be included in the text and where is mentioned the data from genomic analyses.

Reviewer #2 (Remarks to the Author): Expert in CRC/CAC genomics and genetics

Chatila and colleagues analyzed a cohort of IBD patients with colitis-associated cancers by integrating clinical and genomic analysis by studying germline and somatic profiles.

The quality of the data:

Probably, the authors have some trouble with the patients analyzed in the study, I don't understand how many patients were included and analyzed.

Line 73-74

We assembled a series of 174 CAC patients with full clinical data on 144 patients and next-generation sequencing of 166 tumors from 161 patients (Supplementary Table 1, Fig. 1a).

In Table 1 they reported 166 IBD patients, in supplementary 1 reported 166 specimens and in methods section 209 samples.

At the same time, it seems that the percentage reported in the paper was referred to 144 patients (lines 75-80).

They used two different genes panels, both at germline and somatic level.

At the germline level, there were some commonalities between the panels? How many genes were analyzed for each patients?

The results seem not to be carefully interpreted: there are few and poor data about the gene variant interpretation for the APC gene and none for the remaining variations identified in (maybe?) 7 patients. The data about further genes variants were only reported in figure 1, no mention was done in the main text, Why?

Germline analysis was performed on 74 patients, 54% of what?

They reported in lanes 310-311 "Germline sequencing was performed for patients who signed written consent for testing (MSK protocol 12-245) and consisted of a panel of 76 or 88 genes."

What does it mean? that some patients were analyzed for 76 genes while others for 88 genes? Are there some common genes?

Please better define the identified variation in the APC gene, it is I1307K (line 105) or I1301K (line 107)? Please add also "c." for coding DNA sequence, "p." for protein and the class of pathogenicity as reported in you reference n 40.

The level of support for the conclusions:

Lines: 109- 110

"Overall, no germline alteration was statistically significantly enriched among CAC patients".

What was the aim of the study?

Identifying of susceptibility variations (SNP) in IBD by sequencing a genes panel or the identification of possible private pathogenic variations able to trait the IBD-CACs?

In the first hypothesis, no data from healthy controls were available while in the second one, no interpretation of the possible pathogenic variations was shown in the text.

There are many limitations and typos in the paper even when the somatic landscape was analyzed.

It is notorious that the somatic mutation in genes named TP53, KRAS, MYC, PIK3CA, SMAD4, and APC were the most frequently altered genes in several sporadic cancers and in IBD CAC patients.

They performed a targeted sequencing by using 2 different kits, with no news about the gene list, and no news about the performance of the kits etc.

This information was reported in their well-done previous paper published in Gastroenterology in 2016 (Yaeger R et al) performed in a sub-set of 47 patients (included in the current paper; Why here they reported 48 patients? Typo??? Or insufficient attention????).

One person, reading this article thinks about understanding the difference between the two studies, but in the discussion section, there are no comments, differences or, above all, novelties.

Reviewer #3 (Remarks to the Author): Expert in CRC/CAC genomics and genetics

Manuscript “Integrated clinical and genomic analysis of dysplasia and carcinoma identifies driver events and molecular evolution of colitis-associated cancers” by Chatila and others describe clinical and molecular genetic characters of up to 174 patients with colitis associated bowel cancer. This manuscript provides a broad overview of the common features of colitis associated cancer and has a set of patient derived xenograft and organoid experiments supporting the patient based observations. The work seems quite robust, though descriptive. Main weakness of the work is lack of controls, both sporadic CRC tumors and e.g. normal colon organoids (which ought to be Wnt dependent/responsive). In most cases, the lack of controls is ameliorated by referring to prior publications.

Detailed comments:

Lines 73-74 & Supplementary Table 1. The number of patients and samples are somewhat complicated to cipher. The data seems to be collected from something between 174 and 144 patients with some patients with multiple tumors (maybe 166 tumors minus 161 patients i.e. five? Though at line 170 it says six). It would be good to clarify these, even though I can't immediately think of a clear visualisation. Maybe venn diagrams of upset plot.

Related to that, many places in the manuscript state the absolute count of a type of tumors detected, without providing size of the base set for that type. These should be improved.

Lines 104-110. Germline variants. It's not clear how "pathogenic" variants were identified. 14% sounds excessively high proportion unless the patient set is strongly selected for very familial cases or the classification for pathogenicity is very lenient.

On lines 109&110 there is a statement about "no statistically significant enrichment" for germline alteration among CAC patients. How is this conclusion made since the manuscript do not appear to have healthy or non-CAC controls.

Point-by-point Response to Reviewers' Comments:

Reviewer #1 (Remarks to the Author): Expert in colorectal cancer functional genomics, organoids, and mouse models

The manuscript by Chatila et al. contribute to understand how chronic inflammation influences colon cancer development. They combine clinical data, genomic, and germline alterations to mainly suggest that inflammatory conditions promote lesions that occur as genetically independent events. The manuscript is interesting however I find that the data presented lacks certain novelty. I find it is an extended report of what has been published in the topic by the same authors and/or other colleagues in the field. For example:

- Robles et al., Gastroenterology 2016 (whole-exome seq IBD; less number of cases but overlapping findings with TP53, APC and distinct genetic features of CACs compared to sporadic cancers)
- Yaeger et al., Gastroenterology 2016 (same group and analyzed data collected until 2016, focussed on TP53, IDH1, genetic features and CD/UC differences)
- Rajamaki et al., Gastroenterology 2021 (CACs and Wnt pathway, genetic and epigenetic events)

These reports are briefly explained throughout the text in spite of its relevance.

The data: well analysed, well interpreted and conclusions describe the main findings.

The methodology: well described and it is obvious from the report that the group has a strong background in these methodologies.

Major points

1. The authors should make clear what is the novelty compared to previous studies and in the manuscript state what are the samples that were used in the 2016 manuscript. Also, I would have expected a brief paragraph in the introduction to explain the new aim of this study and the novelty compared to the previous one.

Response: We have reorganized the manuscript and included additional data to emphasize our novel observation that dysplastic and cancerous lesions developing in patients with IBD occur as genetically independent events. In support of this observation, the revised manuscript includes new data from an AOM/DSS mouse model of inflammatory colon cancer, in addition to sequencing data we had from a patient who had six synchronous dysplastic/cancerous lesions, 13 patients with paired synchronous dysplasia-cancer specimens, and six patients with multiple IBD-associated primaries. We have added data from an AOM/DSS mouse model of inflammatory colon cancer where mice develop multiple early neoplastic lesions within a period of weeks, after AOM treatment and multiple cycles of colitis induced by DSS administration (new Figure 4a). Using two such mouse models, we have isolated two separate lesions (dysplasia or intramucosal carcinoma) developing in each mouse and performed next-

generation sequencing. The separate lesions were adjacent as they could be encompassed on a single slide and were separately macro-dissected for sequencing. Sequencing of these lesions suggested that they were genetically unrelated; the two lesions from one mouse had one common genomic alteration and the two lesions from the other mouse shared no genomic changes, providing orthogonal support for our observation that multiple lesions developing in patients with IBD are largely genetically independent.

The manuscript has been overall revised to improve readability and we added section headings and subsection headings. We have added to the manuscript Introduction section a paragraph describing the aims of this project and specifically putting these aims within the context of our prior genomic analysis:

Our group and others have previously described recurrent genomic alterations and some of the distinct genomics of CAC⁸⁻¹⁰. In this study, we aimed to map out the genomic steps in the development of these lesions by sequencing dysplastic lesions and, where available, mucosa; evaluating the relationship between dysplasia and cancer or multiple primaries occurring in the same person; and expanding the genomic, genetic, and functional characterization of CAC.

The revised manuscript includes a new figure (Supplementary Figure 1) that shows the flow of patient samples and indicates the number of patients studied for the clinical analyses and the overlap between the clinical and genomic cohorts studied. We have indicated in this diagram where the cases from the 2016 manuscript were included (all 47 included in the genomic cohort and only 28 included in the clinical cohort). We have also added a sentence in the Results section to clearly state that the cases from the 2016 manuscript were further annotated for clinical information and included in this updated series. We have added a tracker to the updated oncoprint in Figure 1b that shows the sequencing assay used (row five).

2. Why do the authors not compare UC and CD data? They did not find major differences? and only the ones reported in Yaeger et al 2016. Or alternatively, are these previous findings confirmed?

Response: The frequencies of all recurrent genomic alterations shown in the oncoprint in Figure 1 were compared between cases arising in the setting of CD and those arising in the setting of UC. The Results section includes the sentence “*PIK3CA* and *IDH1* alterations were significantly enriched in patients with a history of Crohn’s disease versus ulcerative colitis” as these were the only genes found to significantly vary in frequency by IBD subtype. We have clarified in the text that all recurrent genomic alterations were compared in frequency between UC-CAC and CD-CAC and only these two genes varied by IBD subtype. The oncoprint in figure 1 is also sorted by IBD subtype to help visually compare the spectrum of genomic changes in patients with CD versus those with UC.

3. Figure 1 and 2 lacks novelty. However, Figure 3 data are interesting. Can the authors extend the analysis testing PDOs from these samples and drug testing? or alternatively find a way to analyse further the distinct genetic profiles? can they present statistics to proof well the findings?

Response: We have revised Figure 1 to include NGS results from 29 dysplasia samples. This adds novelty to this figure as the spectrum of genomic changes in dysplasia developing in IBD patients is not established and this figure now shows a substantial series of dysplasia too with clinical annotation and deep sequencing results. Regarding the other analyses in this figure, we believe the association of KRAS alteration status with tumor differentiation in CAC (Figure 1c) has not been reported and the deep comparison of copy number changes with sporadic CRC is a new analysis in this manuscript. The clinical data (Figure 1a) and germline analysis (Figure 1d) are important because of the size of our cohort and our ability to look at new relationships. Looking at the clinical literature on CAC, the size of our series stands out and we believe it is important to understand the presentation, histologic features, and metastatic tropisms of CAC to understand the biology and development of this disease.

We appreciate that the reviewer found the data presented on Wnt activation in CAC interesting. These data are part of Figure 2 in the manuscript. We have added statistical comparisons and p-values for the growth curves shown in Figure 2 and Supplementary Figure 2. The analysis of transcriptional profiles (consensus molecular subtypes) involves few cases (n=11), so we couldn't do a statistical comparison. We have expanded the data on Wnt independence of CAC-derived organoids by adding photographs showing the morphology and growth of normal colon organoids in Wnt-/RSPO- conditions (Figure 2c)

4. One of the main challenges is to prove that CACs can be treated according to its signature/profile and that this would be more efficient than when CACs are treated as sporadic tumours. The authors should perform some experiments in this respect. This would give the novelty that is lacking.

Response: We find that colitis-associated dysplasia and cancer have some distinct, potentially targetable alterations compared to sporadic CRC, such as IDH1 mutation or FGFR amplification. We have included functional data (e.g., 2HG production, PDX growth experiments) to support this finding and mention the potential for such targeted approaches in the Discussion section. However, such targetable alterations occur in a minority of CAC cases. We do not have data to nominate specific treatments based on signature/profile in the large majority of CAC but believe that our work, by defining genomic changes in dysplasia and CAC and the relationship between multiple neoplastic lesions in IBD, sets the stage for further studies to develop unique approaches for CAC. We note that such approaches do not even exist right now for sporadic CRC, where tumor signature does not dictate treatment, and only the minority of patients with metastatic disease have treatment choice guided by tumor genomic profile - MSI-H (2-4%), BRAF V600E (8%), HER2 amp (3%) (no approved treatment yet for HER2 amplified CRC but NCCN guidelines include targeted therapies). While in sporadic tumors, Wnt activation seems to drive cancer initiation and maintenance (as shown in PMID 26091037), our work indicates, in agreement with others, that TP53 mutation is the critical event for CAC development in the majority of cases. Further efforts to target TP53 may provide new avenues to treat CAC and better tailor treatments for this disease but are beyond the scope of our study.

Minor points:

1. Histology in Figure 5a should be improved (magnification, arrows showing the major differences...)

Response: We have updated the histology photographs included in the manuscript to show representative areas at higher magnification and added a scale bar for each image.

2. Abstract should describe better the novelty of the study.

Response: Abstract has been revised to more clearly state the novel findings of the study.

3. In general, the manuscript statistics could be improved in the methods section and the p values description. P values should be clear in figures.

Response: We have added p-values to figures or legends where they were previously not included, such as for the growth curves in Figure 2 and Supplementary Figures 2 and 3. We have updated the statistical analysis section in the methods subsection of the manuscript.

4. Genomic analyses were done by FoundationOne in 34 samples and MSK-IMPACT in 132 samples. These methods only reveal hundreds of genes and the limitation should be included in the text and where is mentioned the data from genomic analyses.

Response: We thank the reviewer and appreciate this point. We have noted as a limitation of our study in the Discussion section: "Additionally, we performed primarily targeted sequencing of known cancer genes with whole exome sequencing performed in a subset of cases, so the analysis may have missed important non-cancer genes involved in CAC development." Both the MSK-IMPACT and FoundationOne assays include greater than 300 genes, sequence at high depth, and provide copy-number coverage. We therefore don't believe meaningful oncogenic drivers would be identified beyond these targeted assays.

Reviewer #2 (Remarks to the Author): Expert in CRC/CAC genomics and genetics

Chatila and colleagues analyzed a cohort of IBD patients with colitis-associated cancers by integrating clinical and genomic analysis by studying germline and somatic profiles.

The quality of the data:

Probably, the authors have some trouble with the patients analyzed in the study, I don't understand how many patients were included and analyzed.

Response: We have added a new figure (Supplementary Figure 1) that shows the flow of patient samples and indicates the number of patients studied for the clinical analyses and the overlap between the clinical and genomic cohorts studied. We have added a new subsection at the start of the Results section with the subheading "Study Population" that summarizes the number of patients and samples analyzed. For clarity, we have now distinguished our "clinical

cohort,” which consists of 144 patients with CAC, and our “genomics cohort,” which consists of 161 patients with CAC from whom 166 tumor samples were sequenced.

- Line 73-74: We assembled a series of 174 CAC patients with full clinical data on 144 patients and next-generation sequencing of 166 tumors from 161 patients (Supplementary Table 1, Fig. 1a). In Table 1 they reported 166 IBD patients, in supplementary 1 reported 166 specimens and in methods section 209 samples. At the same time, it seems that the percentage reported in the paper was referred to 144 patients (lines 75-80).

Response: We thank the reviewer for pointing out this confusion. As described above, we have now clarified that 174 CAC patients were studied and the breakdown of cases for the clinical and genomic analyses are updated in a new Supplementary Figure 1 and in the text. Table 1 is shown at the sample level and this is now indicated in the table. It gives the characteristics related to each CAC case sequenced. 166 samples were sequenced from 161 patients so the sample level summary has a slightly higher number than the number of patients. The reason we show clinical data at the sample level is we treated each CAC as a new event and looked, at the time of that cancer development, at clinical characteristics - both related to the patient (duration of IBD up to development of that tumor, etc) and to the tumor (stage, location, presenting symptoms, histologic appearance of adjacent mucosa, etc).

The 144 patients, the denominator for the percentages in the “Clinical features of CAC development” subsection, corresponds to our “clinical cohort” of 144 patients, which is now more clearly described in the Results text and Supplementary Figure 1. The revised manuscript includes section and subsection headings, which should also clarify the population studied and help with readability of the manuscript.

We have reviewed the Methods section of the manuscript and made sure the number of samples are correct. There is no mention of 209 samples in this section.

- They used two different genes panels, both at germline and somatic level. At the germline level, there were some commonalities between the panels? How many genes were analyzed for each patients?

Response: We thank the reviewer for pointing out this confusion. We have now clarified both in the Results and Methods sections that a single assay was used for germline testing that had 76 genes in its first iteration (Version 1, V1) and included 88 genes in its second iteration (Version 2, V2). We added a reference for the gene list (PMID 34133209). The overlap between V1 and V2 of the assay is 84% (14 genes were added and two removed for V2). The germline panel was developed by our molecular biology and genetics groups to capture all known cancer risk genes based on the American College of Medical Genetics and Genomics (ACMG) guidelines and changes from V1 to V2 reflect changes in the current understanding of genes involved in cancer susceptibility.

We have added information on the sequencing assay used as a new bar in the oncoprint in Figure 1b (row 5). References in the Methods section include the gene list for the targeted assays used for somatic mutation testing.

- The results seem not to be carefully interpreted: there are few and poor data about the gene variant interpretation for the APC gene and none for the remaining variations identified in (maybe?) 7 patients.

Response: Thank you for bringing this to our attention and in response to the reviewer's concern, we have substantially expanded the paragraph on germline alterations in collaboration with co-author Dr. Stadler, a GI geneticist. We have included an explanation of the low-penetrance APC I1037K variant finding as well as explanations for the other findings and their relevance to our data. The genes with germline alterations are now listed in the text, in addition to the figure. As discussed below, the frequency of germline variants is similar to that seen in sporadic colorectal cancer. We have added references and more directly made this comparison in the updated text.

- The data about further genes variants were only reported in figure 1, no mention was done in the main text, Why?

Response: As per above, we have expanded the paragraph on germline alterations to describe the gene variants identified. We have added a supplementary table (copied here) that lists the precise germline alterations identified.

Gene affected	Alteration	Number of patients
APC	c.3920T>A (p.Ile1307Lys)	3
PMS2	c.943C>T (p.Arg315*)	1
PMS2	c.137G>T (p.Ser46Ile)	1
FANCA	exon 9-23 deletion	1
DICER1	c.4972delA (p.Thr1658Hisfs*2)	1
ATM	c.7875_7876delinsGC (p.AspAla2625GluPro)	1
FANCC	c.456+4A>T	1
RAD51B	c.321delA (p.Gly108Valfs*12)	1

- Germline analysis was performed on 74 patients, 54% of what?

Response: We have added the denominator to the text.

- They reported in lines 310-311 "Germline sequencing was performed for patients who signed written consent for testing (MSK protocol 12-245) and consisted of a panel of 76 or 88 genes." What does it mean? that some patients were analyzed for 76 genes while others for 88 genes? Are there some common genes?

Response: As mentioned above, we have added information to the Results text and Methods sections to clarify that a single germline assay was used but that the first iteration (Version 1) had 76 genes and the second iteration (Version 2) had a total of 88 genes analyzed. We have clarified the number of patients analyzed with the 76-gene assay (n=15) and the number of patients analyzed with the 88-gene assay (n=59).

- Please better define the identified variation in the APC gene, it is I1307K (line 105) or I1301K (line 107)?

Response: We thank the reviewer for catching this typographical error and have now fixed it to correctly be I1307K.

- Please add also “c.” for coding DNA sequence, “p.” for protein and the class of pathogenicity as reported in your reference n 40.

Response: We have used these notations for the actual alterations listed in new Supplementary Table 2.

- The level of support for the conclusions: Lines: 109-110: “Overall, no germline alteration was statistically significantly enriched among CAC patients”.

Response: To put the germline analysis findings in context, we have now added a reference and discussed the frequency of germline alterations in sporadic CRC. We note in the manuscript: *The frequency of germline alterations was similar to that seen in CRC (PMID 34405229), where analysis of over 1000 patients using the same panel identified a prevalence of germline alterations of 14%, 16%, and 23%, in patients aged 50+ years, 36-49 years, and 14-35 years, respectively, with Lynch syndrome detected in 3%, 4%, and 8% of these groups, respectively.*

- What was the aim of the study? Identifying of susceptibility variations (SNP) in IBD by sequencing a genes panel or the identification of possible private pathogenic variations able to trait the IBD-CACs?

Response: The aim of this study was to identify the genomic steps underlying CAC development. Our central findings are that neoplastic lesions developing in patients with IBD occur largely as independent genetic events and do not result from a field-effect of inflammation-induced genomic alterations and that the resultant tumors have experienced a lineage shift away from Wnt signaling. The aim of the germline analysis of patients with CAC was to determine if known cancer predisposing genes, once highly penetrant syndromes such as Lynch syndrome are excluded, are enriched in patients who have IBD and then develop cancer. Prior germline analyses (PMID 34347074) have looked at a smaller cohort of IBD patients and used a smaller panel for analysis, so our expanded data clarify the frequency of germline alterations in patients with CAC.

- In the first hypothesis, no data from healthy controls were available while in the second one, no interpretation of the possible pathogenic variations was shown in the text.

Response: Our goal is to understand the molecular steps underlying CAC development. The germline analysis included in our manuscript is meant to study if patients with IBD who develop CAC are predisposed to cancer because of known cancer-related germline alterations. This would clarify if additional germline events underlie the development of these tumors. An example would be if many patients had germline BRCA mutations, this would indicate that BRCA alterations commonly underlie the development of CAC and act as an early molecular step. In our analysis, we did not find a high frequency of germline alterations (neither relative to sporadic CRC or other GI cancers) suggesting that germline changes do not drive the molecular pathogenesis of CAC. We have now added additional interpretation of the identified germline alterations to the manuscript text to better elucidate any potential relevance to CAC carcinogenesis.

- There are many limitations and typos in the paper even when the somatic landscape was analyzed.

Response: The manuscript was carefully reviewed and any typos corrected.

- It is notorious that the somatic mutation in genes named TP53, KRAS, MYC, PIK3CA, SMAD4, and APC were the most frequently altered genes in several sporadic cancers and in IBD CAC patients. They performed a targeted sequencing by using 2 different kits, with no news about the gene list, and no news about the performance of the kits etc.

Response: The MSK-IMPACT and FoundationOne assays have been extensively described and used to characterize tens of thousands of tumors with numerous publications reporting these results. Both the MSK-IMPACT and FoundationOne assays include greater than 300 genes, sequence at high depth, and provide copy-number coverage. We therefore don't believe meaningful oncogenic drivers would be identified beyond these targeted assays. Both assays are FDA-approved for tumor next generation sequencing and their gene list is publicly available. References in the Methods section include the gene list for each assay.

- This information was reported in their well-done previous paper published in Gastroenterology in 2016 (Yaeger R et al) performed in a sub-set of 47 patients (included in the current paper; Why here they reported 48 patients? Typo??? Or insufficient attention????). One person, reading this article thinks about understanding the difference between the two studies, but in the discussion section, there are no comments, differences or, above all, novelties.

Response: We have added to the manuscript Introduction section a paragraph describing the aims of this project and specifically putting these aims within the context of our prior genomic analysis:

Our group and others have previously described recurrent genomic alterations and some of the distinct genomics of CAC⁸⁻¹⁰. In this study, we aimed to map out the genomic steps in the development of these lesions by sequencing dysplastic lesions and, where available, mucosa; evaluating the relationship between dysplasia and cancer or multiple primaries occurring in the same person; and expanding the genomic, genetic, and functional characterization of CAC. We have also addressed the additional knowledge gained from this study in the first paragraph of the Discussion section and expanded the discussion of the implications of our new findings in this section. We have corrected the number of cases from the prior study to 47.

Reviewer #3 (Remarks to the Author): Expert in CRC/CAC genomics and genetics

Manuscript “Integrated clinical and genomic analysis of dysplasia and carcinoma identifies driver events and molecular evolution of colitis-associated cancers” by Chatila and others describe clinical and molecular genetic characters of up to 174 patients with colitis associated bowel cancer. This manuscript provides a broad overview of the common features of colitis associated cancer and has a set of patient derived xenograft and organoid experiments supporting the patient based observations. The work seems quite robust, though descriptive.

Response: We thank the reviewer for these comments.

Main weakness of the work is lack of controls, both sporadic CRC tumors and e.g. normal colon organoids (which ought to be Wnt dependent/responsive). In most cases, the lack of controls is ameliorated by referring to prior publications.

Response: In the revised manuscript, we have added photographs of normal colon organoids grown over time without wnt/r-spondin supplementation for comparison to CAC organoids (Figure 2c). The wild-type normal colon organoids are dependent on exogenous wnt and r-spondin for survival. Withdrawal of wnt and r-spondin from the organoid culture media leads to a loss of epithelial architecture and organoid cell death by day 7. In contrast, the CAC organoids continue to proliferate, unaffected by the wnt deprivation. Normal colon organoids are also included in the growth curves in Figure 2f, and we have now added a statistical comparison of growth to the figure legend. We have clarified in the legend that normal colon organoids are shown. We have added to the revised manuscript text and references describing the frequency of germline alterations in sporadic CRC to put our results for CAC patients in context.

Detailed comments:

- Lines 73-74 & Supplementary Table 1. The number of patients and samples are somewhat complicated to cipher. The data seems to be collected from something between 174 and 144 patients with some patients with multiple tumors (maybe 166 tumors minus 161 patients i.e. five? Though at line 170 it says six). It would be good to clarify these, even though I can't immediately think of a clear visualisation. Maybe venn diagrams of upset plot.

Response: We appreciate this point. The revised manuscript includes a new figure (Supplementary Figure 1) that shows the flow of patient samples and indicates the number of patients studied for the clinical analyses and the overlap between the clinical and genomic cohorts studied. We have added a new subsection at the start of the Results section with the subheading “Study Population” that summarizes the number of patients and samples analyzed. For clarity, we have now distinguished our “clinical cohort,” which consists of 144 patients with CAC, and our “genomics cohort,” which consists of 161 patients with CAC from whom 166 tumor samples were sequenced.

- Related to that, many places in the manuscript state the absolute count of a type of tumors detected, without providing size of the base set for that type. These should be improved.

Response: We have updated the Results section to indicate the denominator for each analysis and clarified where studies are reported for the “clinical cohort” and where they are reported for the “genomic cohort.”

- Lines 104-110. Germline variants. It’s not clear how “pathogenic” variants were identified. 14% sounds excessively high proportion unless the patient set is strongly selected for very familial cases or the classification for pathogenicity is very lenient.

Response: We note in the Methods section that pathogenic variants are identified based on “American College of Medical Genetics and Genomics (ACMG) guidelines⁴⁰ by a clinical molecular geneticist or molecular pathologist.” In the revised manuscript, we have added information about frequency of germline alterations in sporadic CRC to put the proportion of germline changes in patients with CAC in context. We have added a new supplementary table that lists the germline alterations identified (Supplementary Table 2). We note in the revised manuscript that a previous, smaller series (n=25 CAC patients) found that 24% of patients carried pathogenic or likely pathogenic germline variants (PMID 34347074).

- On lines 109&110 there is a statement about “no statistically significant enrichment” for germline alteration among CAC patients. How is this conclusion made since the manuscript do not appear to have healthy or non-CAC controls.

Response: We have revised the germline results paragraph extensively in response to the reviewers’ comments. We now provide information from a reference population tested with the same germline genetic assay: *The frequency of germline alterations was similar to that seen in CRC (PMID 34405229), where analysis of over 1000 patients using the same panel identified a prevalence of germline alterations of 14%, 16%, and 23%, in patients aged 50+ years, 36-49 years, and 14-35 years, respectively, with Lynch syndrome detected in 3%, 4%, and 8% of these groups, respectively.*

REVIEWERS' COMMENTS

Reviewer #1 (Remarks to the Author):

Thanks for the clarifications.

Most of my concerns have been answered and in my opinion the manuscript can be accepted to be published in Nat Comm.

Reviewer #2 (Remarks to the Author):

Please, add in "Table S2. Germline alterations detected in patients with CAC" another column describing the pathogenicity classes according to IARC recommendations.

Reviewer #3 (Remarks to the Author):

The authors have responded to my review satisfactorily and I have no further comments.

Point-by-point Response to Reviewers' Comments:

Reviewer #1 (Remarks to the Author):

Thanks for the clarifications.

Most of my concerns have been answered and in my opinion the manuscript can be accepted to be published in Nat Comm.

Reviewer #2 (Remarks to the Author):

Please, add in "Table S2. Germline alterations detected in patients with CAC" another column describing the pathogenicity classes according to IARC recommendations.

Response: This table has been updated as suggested to include pathogenicity classes noted according to ACMG and to IARC recommendations.

Reviewer #3 (Remarks to the Author):

The authors have responded to my review satisfactorily and I have no further comments.